# H89 Reverses Multidrug Resistance in Colorectal Cancer by Inhibiting the ATPase Activity of ABCB1

**DOI:** 10.3390/biomedicines13122869

**Published:** 2025-11-25

**Authors:** Wei-Jing Liu, Yi-Yao Shan, Huan Wang, Yu-Meng Xiong, Le-Yao Shi, Xiao-Peng Song, Min Li, Ke He, Jia Huang, Zhi Shi

**Affiliations:** 1Cancer Minimally Invasive Therapies Centre, Guangdong Second Provincial General Hospital, Department of Cell Biology & Institute of Biomedicine, Guangdong Provincial Biotechnology & Engineering Technology Research Center, Guangdong Provincial Key Laboratory of Bioengineering Medicine, Guangdong Basic Research Center of Excellence for Natural Bioactive Molecules and Discovery of Innovative Drugs, Genomic Medicine Engineering Research Center of Ministry of Education, MOE Key Laboratory of Tumor Molecular Biology, National Engineering Research Center of Genetic Medicine, State Key Laboratory of Bioactive Molecules and Druggability Assessment, College of Life Science and Technology, Jinan University, Guangzhou 510632, China; lwj2023@stu2023.jnu.edu.cn (W.-J.L.); yiyaoshan@stu2021.jnu.edu.cn (Y.-Y.S.); ymzxx@stu2022.jnu.edu.cn (Y.-M.X.); songxp@stu2023.jnu.edu.cn (X.-P.S.); lixiyu@stu2023.jun.edu.cn (M.L.); heke8@mail3.sysu.edu.cn (K.H.); 2Department of Gynecology, The Third Affiliated Hospital of Sun Yat-sen University, Guangzhou 510630, China; wangh28@mail.sysu.edu.cn; 3International Department, the Affiliated High School of South China Normal University, Guangzhou 510630, China; shily.lily2024@gdhfi.com; 4Reproductive Medicine Center, The First Affiliated Hospital, Sun Yat-sen University, Guangzhou 510080, China

**Keywords:** colorectal cancer, multidrug resistance, ABCB1, H89, ATPase

## Abstract

**Background:** Multidrug resistance (MDR) remains a major obstacle in cancer chemotherapy, and overexpression of ABCB1 plays a critical role in the pathogenesis of MDR. Despite decades of research, significant clinical progress in the development of ABCB1 inhibitors has yet to be achieved. The small-molecule H89 is originally identified as an inhibitor of protein kinase A (PKA), but it also exhibits various functions unrelated to the PKA. This study investigates H89 as a novel ABCB1-inhibitor to reverse MDR in colorectal cancer (CRC). **Methods:** Cytotoxicity assays were performed on ABCB1-overexpressing MDR cell line HCT-8/V and parental CRC cell line HCT-8. Drug accumulation was quantified via flow cytometry, and cell cycle effects were analyzed using propidium iodide staining. The ATPase activity of ABCB1 was detected using an ATPase activity assay kit. Molecular docking utilized the ABCB1 crystal structure. **Results:** Both 3 μM and 10 μM H89 significantly reverses resistance to two ABCB1 substrate drugs (doxorubicin and vincristine) in HCT-8/V cells in a dose-dependent manner, with no such effect observed inHCT-8 cells. The combination of H89 and doxorubicin or vincristine resulted in a significant increase in the proportion ofHCT-8/Vcells in the sub-G1 and G2/M phases. Further mechanistic studies reveal that H89 exerts its effect by inhibiting the drug efflux function of ABCB1, thereby increasing the intracellular accumulation of the substrate drugs and reversing multidrug resistance. Furthermore, H89 did not alter the expression of ABCB1. H89 effectively inhibited the ATPase activity of ABCB1. Molecular docking simulations revealed the binding mode of H89 with ABCB1. **Conclusions:** The combination of H89 with ABCB1 substrate drugs significantly reverses multidrug resistance in colorectal cancer. These findings provide a strong theoretical and experimental foundation for the development of novel MDR-reversing agents targeting ABCB1.

## 1. Introduction

Colorectal cancer (CRC) is one of the three most common types of cancer [1]. According to the latest reports, the global number of new cases has reached 1.926 million, ranking among the top three and accounting for 9.3% of the total global incidence of new cancers [2]. A growing amount of evidence supports that, in addition to environment and genetic factors, early-life exposures have also emerged as a significant risk factor [3]. With advances in medical technology and deeper insights into the biological characteristics of CRC, treatment strategies for CRC have continuously evolved and improved, leading to the establishment of a comprehensive treatment model centered on surgery, coupled with chemotherapy, radiotherapy, targeted therapy, immunotherapy, and other modalities. The most frequent chemotherapeutic agents are 5-fluorouracil (5-FU), oxaliplatin (OX), and irinotecan (IRI). 5-FU, a key chemotherapeutic drug for CRC, is a pyrimidine analog functioning to inhibit DNA synthesis [4]. Capecitabine (CAP), an oral prodrug of 5-FU, is used as adjuvant therapy [5]. Oxaliplatin is a platinum-based antineoplastic agent commonly employed in the treatment of advanced CRC; as an alkylating compound, it suppresses DNA synthesis [6]. Irinotecan, frequently used for metastatic CRC, acts as a topoisomerase I inhibitor, functioning to block DNA replication and transcription [7]. Chemotherapy is typically administered as a combination that comprises two to three chemotherapeutic drugs, such as 5-FU+OX, 5-FU+IRI, or CAP+OX [8]. However, most CRC patients who complete treatment will experience recurrence, and nearly all patients will develop drug resistance [9].

Multidrug resistance (MDR) refers to cancer cells reduced sensitivity to chemotherapy, which ultimately induces cross-resistance to multiple distinct drugs [10]. MDR mechanisms are multifaceted. One of the most widely recognized frameworks categorizing them into three types: drug-dependent, target-dependent, and drug/target independent types. The first two categories generally result from abnormal drug efflux or dysregulation of drug targets. In contrast, drug/target-independent MDR arises from target desensitization induced by alterations in genetics or cellular signaling pathways [11,12,13]. Among these mechanisms, the overexpression of transporters belonging to ATP-binding cassette (ABC) subfamily is critical, which can utilize ATP-derived energy to efflux anticancer drugs out of cells [14,15,16].

ABCB1 (ATP-binding cassette subfamily B member 1), which belongs to the ABC transporter family, is also called P-glycoprotein (P-gp). It has been extensively research as a potential target for reversing chemotherapeutic resistance [17,18]. Human ABCB1 is encoded by *MDR1* gene, which is located on chromosome 7q21. Structurally, *MDR1* gene contains two promoters that differ in function [19]. Literature indicates that the transcript arising from the upstream promoter contains 29 exons, andthis form ismore frequently produced in drug-resistant cancer cells. In contrast, the transcript from the downstream promoter contains 28 exons and is typically expressed in normal cells [20,21]. Structurally, ABCB1 is a symmetric protein, with each side composed of one transmembrane domain (TMD) and one nucleotide-binding domain (NBD) [22]. The TMDs form a drug-binding pocket, which is primary responsibility for recognizing and transporting substrates. The NBDs can bind and hydrolyze ATP, converting chemical energy into conformational changes that promote substrates entering the binding pocket [23,24]. Widely present in the colon, small intestine, and proximal tubules of the kidney [25], ABCB1 also play a vital role in the absorption, metabolism, and efflux of various drugs, thereby influencing their distribution in barrier tissues [26]. On the other hand, the overexpression of ABCB1 in tumors leads to the progression of MDR in cancer cells against many common drugs that serve as ABCB1 substrates, such as doxorubicin, paclitaxel and vincristine [27,28,29], particularly in CRC [30,31,32,33,34,35].

To overcome MDR mediated by ABCB1, a multitude of small-molecule inhibitors targeting ABCB1 have been developed, including verapamil, valspodar, and zosuquidar [36,37]. These ABCB1 inhibitors are primarily classified into competitive and non-competitive inhibitors. Competitive inhibitors directly bind to ABCB1’s drug-binding pocket and contend with substrate drugs for the active site, thus reducing the efflux of substrate drugs. Non-competitive inhibitors include: (1) allosteric modulators that alter the active conformation of ABCB1; (2) agents that interfere with ATP binding; and (3) compounds that interact with the membrane and disrupt its lipid environment [38,39]. Despite decades of efforts, although cell-based and animal studies have reported that ABCB1 inhibition exhibits anticancer effects, none of these compounds has achieved success in clinical applications [40]. Another strategy to overcome ABCB1 induced MDR is to inhibit its expression [41]. Researchers have developed gene therapies, which have demonstrated that downregulating ABCB1 expression can also successfully reverse MDR in cancer cells [42,43]. Additionally, various small-molecule drugs and natural productscan work synergistically with common chemotherapeutic agents to exert anti-cancer effects [44,45].

Protein kinase A (PKA) inhibitor H89 exerts inhibitory impacts on the kinase activity of PKA through binding in a competitive manner to the ATP-binding domain [46]. Since the PKA pathway is centrally involved in cellular signal transduction, H89 typically counteracts the cellular responses induced by enhanced PKA activity [47,48]. A rising body of research has revealed that H89 also exhibits additional effects independent of the PKA pathway; for instance, it can be combined with antineoplastic agents to enhance therapeutic efficacy [49], induce apoptosis [50], and trigger antitumor immune responses [51]. In this study, we are the first to propose that H89 has a novel function of inhibiting the ATPase activity of ABCB1, which provides new insights for developing more effective treatment modalities.

## 2. Materials and Methods

### 2.1. ReagentsandCell Culture

H89 dihydrochloride (#460618), 3-(4,5-dimethylthiazol-yl)-2,5-diphenyl-tetrazolium bromide (MTT) (#Q108115), doxorubicin hydrochloride (#A603456-0025), verapamil hydrochloride (#V4629), vincristine sulfate (#2068-78-2), oxaliplatin (#A8648), and rhodamine 123 (#83702) were purchased from TargetMol Chemicals Inc. (Shanghai, China), Dibo Biotechnology Co. (Shanghai, China), Bio Basic Inc. (Shanghai, China), Sigma-Aldrich (Shanghai, China), Aikon Biopharmaceutical R&D Co. (Nanjing, Jiangsu, China), APExBIO Technology LLC (Houston, TX, USA), and Sigma-Aldrich Trading Co. (Shanghai, China). Anti-ABCB1 antibody (#sc-8313) and anti-β-tubulin antibody (#30302ES60) were ordered from Santa Cruz Biotechnology Co. (Santa Cruz, CA, USA) and Yeasen Biotechnology Co. (Shanghai, China). The HCT-8/V, a multidrug-resistant human colorectal cancer cell model, was created by gradually exposing the HCT-8 cells to vincristine and progressively raising the drug concentrations [52]. All cell lines were grown in Dulbecco’s modified Eagle’s medium supplemented with 10% CellMax fetal bovine serum (#SA301.02.V) from Lanzhou Minhai Bio-Engineering Co. (Lanzhou, China) and maintained in a humidified incubator at 37 °C with 5% CO_2_.

### 2.2. Cytotoxicity Assay

Cells were seeded into 96-well plates at a concentration of 5 × 10^3^ cells per well. Following cell attachment, the respective agents were added for a 72-h incubation period. Upon treatment completion, 500 μg/mL MTTwas added and the plates were incubated for 4 h. Subsequently, the supernatant was carefully aspirated, and 50 μL DMSO was added. Absorbance at 570 nm was measured via BioTek Synergy H1 plate reader (Agilent Technologies, Santa Clara, CA, USA). IC_50_ values were calculated with GraphPad Prism 8.0.

### 2.3. Cell Cycle Assay

Cells were seeded in 6-well plates at a concentration of 2.5 × 10^5^ per well. After the cells had adhered to the plate, the corresponding agents were added for a 24-h treatment. Cells were then collected and resuspended in pre-chilled 70% ethanol for 30 min. After centrifugation at 300× *g* for 5 min, they were gently resuspended in 200 μL staining solution. The mixture was then incubated in the darkat room temperature for 15 min. The staining buffer contained 50 μg/mL propidium iodide, 0.1% sodium citrate, 0.1% Triton X-100, and 100 μg/mL RNase A. A Beckman CytoFLEX flow cytometer (Brea, CA, USA) was utilized for fluorescent signals detectionat excitation 480 nm and emission 585 nm. Experimental data were analyzed with ModFit LT 5.

### 2.4. Drug Accumulation Assay

Cells were plated in 6-well plates at 5 × 10^5^ cells/well, then pretreated with H89 or verapamil for 1 h. One hour later, 10 μM doxorubicin or rhodamine 123 was added for an additional 2-h treatment. Fluorescence imaging was performed using an Olympus CKX53 microscope (Tokyo, Japan) to assess the intracellular accumulation of the substrate drugs. Subsequently, cells were harvested, and fluorescent quantitative analysis was conducted using a Beckman CytoFLEX flow cytometer. The data were processed via FlowJo v10.8 software.

### 2.5. Western Blot Assay

Cells were placed in pre-cooled lysis buffer (containing 0.1% sodium dodecyl sulfate, 0.5% sodium deoxycholate, 1% Nonidet P-40, 10 ng/mL phenylmethanesulfonyl fluoride and 0.03% aprotinin) and lysed on ice for 30 min. After centrifugation (14,000× *g*, 10 min), the clear supernatant was collected. Proteins were separated by 10% sodium dodecyl sulfatepolyacrylamidegel electrophoresis and subsequently transferred to polyvinylidene difluoride membranes. The membranes were blocked with 5% BSA at room temperature for 1 h, followed by sequential incubation with primary antibodies specific to the target protein and horseradish peroxidase-conjugated secondary antibodies. Ultimately, signal detection was conducted via chemiluminescence using a gel imaging system from Bio-Rad Laboratories.

### 2.6. ABCB1 ATPase Activity Assay

The detection was performed using anABCB1 ATPase Activity Assay Kit (#V3601, Promega Corporation, Madison, WI, USA). In a 96-well plate, the membrane protein ABCB1 was co-incubated with different concentrations of H89 for 5 min. Subsequently, 5 mM Mg^2+^-ATP was introduced to initiate the ATPase activity reaction. The mixture was then incubated at 37 °C for 40 min before adding ATP Detection Buffer to trigger luminescence. Finally, the emitted light was measured with a multi-functional microplate reader manufactured by Gene Company Limited (Chai Wan, Hong Kong).

### 2.7. Molecular Docking Analysis

The human ABCB1 protein structure (PDB ID: 4Q9H) was retrieved from the Protein Data Bank (PDB). The structure of H89 (CID: 5702541) was obtained from Pub-Chem. Molecular docking simulations of H89 or ATP with ABCB1 were executed with AutoDock4.2 software and were optimized by removing water molecules or heteroatoms and adding hydrogen atoms. The grid box centers were set to 73.104, −3.752, −5.073 and 30.521, 26.518, 32.519, respectively. The grid box sizes were set to 47, 43, 51 and 40, 49, 53, respectively, to ensure full coverage of the active sites. Each molecular docking simulation was run 5 times, resulting in RMSD values of 0.531 and 0.487, respectively. Finally, the binding conformations were displayed in Python 3.7, PyMOL 1.5.6, and OpenBabel 3.1.1.

### 2.8. Statistical Analysis

Statistical significance was analyzed using Student’s *t*-test and One-WayANOVA in GraphPad Prism 9 software. The resultant values were described as the mean ± SD for the indicated number of independent experiments. * *p* < 0.05, ** *p* < 0.01, *** *p* < 0.001 were considered statistically significant.

## 3. Results

### 3.1. H89 Reverses ABCB1-Mediated MDR in CRC Cells

To investigate the effect of H89 (Figure 1A) on ABCB1-mediated MDR in CRC cells, we initially evaluated the effect of H89 on colorectal cancer cell viability through cytotoxicity assays. The data revealed that at concentrations below 30 μM, H89 exhibited low cytotoxicity to both HCT-8 and HCT-8/V cells (Figure 1B). Based on this result, 3 μM and 10 μM concentrations of H89 were selected for subsequent experiments. Compared with HCT-8, HCT-8/V cells showed significantly higher resistance to ABCB1 substrate drugs (doxorubicin and vincristine), reaching 19.94-fold and 1200-fold, respectively (Table 1). Both 3 μM and 10 μM H89 could effectively reverse HCT-8/V cell resistance to doxorubicin and vincristine dose-dependently, whereas the similar effect was absent in HCT-8 cells (Figure 1C). Among them, the reversal effect of 10 μM H89 exceeded that of the ABCB1-specific inhibitor verapamil at the same concentration. Notably, neither H89 nor verapamil enhanced the sensitivity of these two colorectal cancer cells to oxaliplatin (non-ABCB1 substrate drug).

### 3.2. The Combination of H89 with Doxorubicin or Vincristine Induces Cell Cycle Arrest

To investigate the effects of H89 combined with doxorubicin and vincristine on two colorectal cancer cell lines, we examined changes in the cellular division cycle after drug treatment using flow cytometry. As shown in Figure 2A,B, in HCT-8/V cells, compared to the single-drug treatment groups, the H89-doxorubicin combination group exhibited a significant increase in the proportions of cells in the sub-G1 and G2/M phases. Meanwhile, the H89-vincristine combination group showed a significant increase in the proportion of sub-G1 phase cells and a drop in G2/M phase cells. However, this phenomenon was not observed in HCT-8 cells.

### 3.3. H89 Enhances theIntracellular Accumulation of Substrate Drugs in HCT-8/V Cells

To verify whether H89 can affect the efflux function of ABCB1 on its substrate drugs, we performed drug accumulation assays targeting the ABCB1 substrate drugs (rhodamine 123 and doxorubicin). The results revealed that compared to the parental cells, the resistant cells exhibited the lower intracellular concentrations of doxorubicin and rhodamine 123. Furthermore, H89 elevated doxorubicin and rhodamine 123 levels in HCT-8/V cells dose-dependently. At the same concentration of 10 μM, H89 achieved or even exceeded the inhibitory effect of verapamil. Treatment with 30 μM H89 increased the intracellular rhodamine 123 levels in HCT-8/V cells to a level comparable to that in HCT-8 cells. Notably, the similar effect of H89 was not observed in HCT-8 cells (Figure 3A–F).

### 3.4. H89 Inhibits the ATPase Activityof ABCB1 and Occupies Its ATP-Binding Pocket

To determine whether H89 affects ABCB1 protein expression, we treated HCT-8/V cells with 10 μM and 30 μM H89 for 0.5, 3, and 72 h, respectively. The results indicated that H89 did not alter ABCB1 protein level relative to the loading control β-tubulin (Figure 4A). Furthermore, ATPase activity assays revealed that H89 inhibited the ATPase activity of ABCB1 in a dose-dependent manner (Figure 4B), with an IC_50_ of 0.246 μM. We hypothesized that H89 might competitively bind to the ATP-binding pocket, thereby concurrently attenuation of both the ATPase activity and efflux function of ABCB1. This hypothesis was verified by molecular docking simulations. As shown in Figure 4C,E, H89 docked favorably into the ATP-binding site, witha high binding affinity of −8.57 kcal/mol. We also evaluated the affinity of ATP for the ABCB1 ATP-binding pocket and found that its binding energy was only −4.51 kcal/mol, which was significantly higherthan that of H89 (Figure 4D,F). These results indicate that H89 inhibits the ATPase activity of ABCB1 and hinders substrate drug efflux by competitively occupying its ATP-binding domain.

## 4. Discussion

Over the past several decades, CRC has emerged as a leading global malignancy. Despite progress in treatment, CRC still exhibits a high mortality rate. Data indicate that over 90% of cancer-related deathsare associated with MDR. The development of MDR following chemotherapyinvolves multiple mechanisms, including heightened drug-pumping, genetic elements, excessive secretion of growth factors, augmented DNA repair ability, and upregulated drug metabolic processes [53,54,55,56]. Among these factors, the upregulation of drug efflux systems significantly contributes to the development of chemoresistance [57,58]. For instance, the overexpression of ABC superfamily transporters markedly impairs the therapeutic efficacy of chemotherapeutic agents such as doxorubicin, cisplatin, and 5-fluorouracil (5-FU) [59,60,61,62]. The binding of substrates to ABC transporters inducesconformational changes.Subsequently, these transporters exploit energy fromATP hydrolysis to expel the bound substrate drugs out of cells [63,64].

As a prototypical member of the ABC transporter family, ABCB1 has been definitively implicated in chemoresistance across multiple cancers. Inhibiting the expression or function of ABCB1 has therefore become a key strategy for reversing MDR, as evidenced by the development of various ABCB1 inhibitors. However, the clinical efficacy of these inhibitors has consistently been suboptimal. Our study demonstrates that the small-molecule drug H89 could reverse MDR to ABCB1 substrate drugs in CRC cells, with this synergistic effect strengthening in a concentration-dependent manner. We first verified that H89 exhibited minimal cytotoxicity in both the parental CRC cell line HCT-8 and the ABCB1-overexpressing MDR cell line HCT-8/V at concentrations below 30 μM. Cytotoxicity assays showed that H89 potentiated the efficacy of ABCB1 substrate drugs in HCT-8/Vcells in a concentration-dependent manner, with no such effect in HCT-8 cells. Furthermore, neither H89 nor verapamil (a positive control ABCB1 inhibitor) altered the sensitivity of either cell line to oxaliplatin (non-ABCB1 substrate drug). Consistent with these findings, drug accumulation experiments demonstrated that H89 enhanced the intracellular accumulation of doxorubicin and rhodamine 123 (both ABCB1 substrates) in the MDR subline, but not in the parental cells, in aconcentration-dependent manner. Notably, at the same concentration, the MDR-reversing efficacy of H89 was comparable to orslightly superior to that of verapamil.

To elucidate the mechanism underlying H89-mediated reversal of ABCB1-dependent MDR, we first employed Western blot analysis, which ruled out the possibility that H89 modulates ABCB1 protein expression. Given that ABCB1 functions as an efflux pump, its activity is dependent on its intrinsic ATPase activity. We investigated whether H89 affects the ATPase activity of ABCB1. The results indicated that H89 suppressedthe ATPase activity of ABCB1 in a dose-dependent manner, with an IC_50_ value of 0.246 μM. It is known that H89 inhibits PKA by competing for its ATP-binding site [65]. We therefore hypothesized is that H89 inhibits the ATPase activity of ABCB1 through a similar mechanism. Previous studies have suggested that hydrogen bonds between ABCB1 substrates and key amino acids can reduce its ATPase activity [23]. Therefore, we conducted molecular docking simulations to analyze the binding of H89 to ABCB1. The data revealed a binding energy of −8.57 kcal/mol for H89 at the ATP-binding site of ABCB1, which is significantly more favorable than the binding energy of ATP itself (−4.51 kcal/mol) for the same site. This further confirmed the strong competitive binding of H89 for the ATP-binding site of ABCB1.

This study is the first to identify H89 as a promising agent that antagonizes ABCB1-mediated MDR in CRC. Notably, independent of its intrinsic PKA inhibitory activity, H89 cantrigger reactive oxygen species. When combined with glyceryl trinitrate, H89 can enhance the apoptotic response of CRC cells [49]. H89 is also capable of inhibiting cell growth through a novel pathway independent of targeting PKA-Cα, MSK1, or S6 kinase. The detailed mechanism primarily involves increased CARP-1 phosphorylation, its interaction with transcriptional co-activator, and c-myc suppression [66]. Additionally, the combined therapy of H89 and tetrandrine enhanced anticancer activity in vitro and in vivo by inducing apoptosis and autophagy, without exerting adverse effects on normal cell function [67]. In pancreatic cancer cells, however, the suppression of cell growth and the induction of apoptosis by H89 and PKI (another PKA inhibitor) are mediated through the suppression of PKA [68]. Similarly, owing to its inhibitory effects on multiple kinases, the combination of H89 with recombinant immunotoxins significantly enhances immunotoxinactivity against acute lymphoblastic leukemia cell lines [69]. It is noteworthy that staurosporine analogs, which are specific inhibitors of protein kinase C, can also reverse MDR in diseases such as breast cancer [70], colon adenocarcinoma [71], and leukemia [72]. These findings highlight the paradigms of regulating kinase signaling pathways for indirect intervention or direct intervention independent of kinase pathways. H89, which possesses dual effects of inhibiting the PKA pathway and directly targeting ABCB1, expands the applicability of MDR reversal and confers certain potential advantages for clinical translation. In CRC, oxaliplatin and other platinum drugs serve as core therapeutic agents, though they are generally regarded as weak ABCB1 substrates or non-ABCB1 substrates. Consistent with this, our experiments confirmed that H89 does not alter the sensitivity of drug-resistant cells to oxaliplatin. This indicates that H89 exhibits clear substrate selectivity in its inhibitory effect on ABCB1. Given that H89 acts as a bifunctional MDR reverser with both “direct ABCB1 inhibition” and “signaling pathway regulation” properties, we propose that the combined application of H89 and platinum drugs via nano-delivery systems holds considerable feasibility.

Understanding the complex mechanisms underlying MDR remains a key focus for developing novel cancer therapeutic strategies. This will likely involve combinations of targeted inhibitors to archive multifunctional anticancer effects and optimize therapeutic efficacy [73,74]. Our study suggests that H89 can effectively inhibit the ATPase activity of ABCB1. Future research should focus on the detailed elucidation of the binding mechanism between H89 and ABCB1, laying the foundation for structural optimization or the design of highly selective derivatives. Furthermore, when exploring the combination regimens of H89 with ABCB1 substrate drugs, the optimal dose ratio and administration sequence should be determined to maximize MDR reversal and minimize toxicity.

## 5. Conclusions

In summary, our findings demonstrate that H89 reverses multidrug resistance in colorectal cancer by inhibiting the ATPase activity of ABCB1. In the context of chemical resistance reversal strategies, the combination of H89 with ABCB1 substrate drugs is a promising approachto overcome ABCB1-mediated MDR.

## Figures and Tables

**Figure 1 biomedicines-13-02869-f001:**
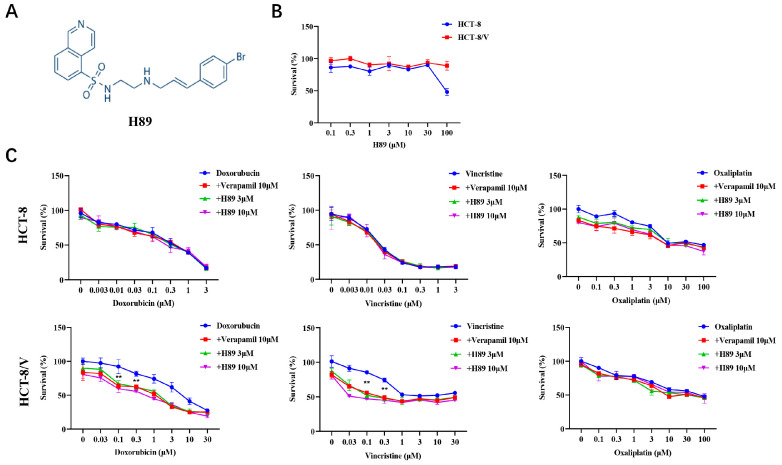
H89 reverses ABCB1-mediated MDR in CRC cells. (**A**) The chemical structure of H89. (**B**,**C**) Cells were treated with the indicated drugs for 72 h and detected by MTT assay. Representative cell viability curves are shown. ** *p* < 0.01 compared to the corresponding control groups. Data are presented as mean ± SD (*n* = 3).

**Figure 2 biomedicines-13-02869-f002:**
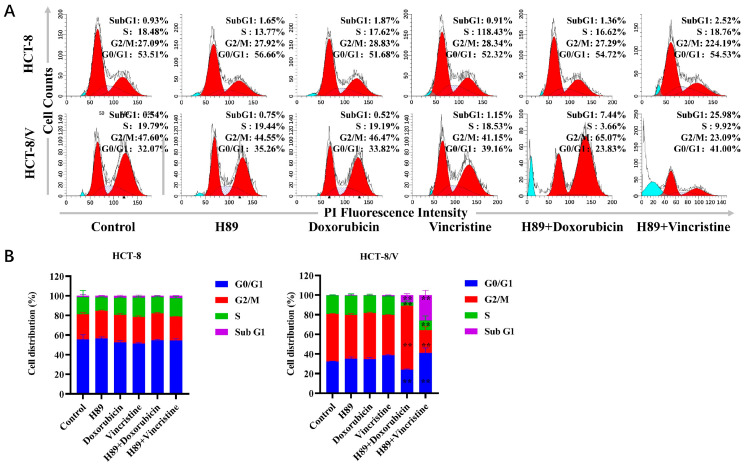
The combination of H89 with doxorubicin or vincristine induces cell cycle arrest. Cells were treated with the indicated reagents for 24 h, and the cell cycle distribution was analyzed by flow cytometry using propidium iodide (PI) staining. The concentrations of each agent were as follows: 10 μM H89, 0.03 μM doxorubicin and 0.003 μM vincristine for HCT-8 cells; 10 μM H89, 0.1 μM doxorubicin and 0.1 μM vincristine for HCT-8/V cells. (**A**) Representative histograms. Bright blue indicates Sub G1 period, and red indicates G0/G1-G2/M period. (**B**) Representative quantitative data. ** *p* < 0.01 compared to the corresponding control groups. Data are presented as the mean ± SD (*n* = 3).

**Figure 3 biomedicines-13-02869-f003:**
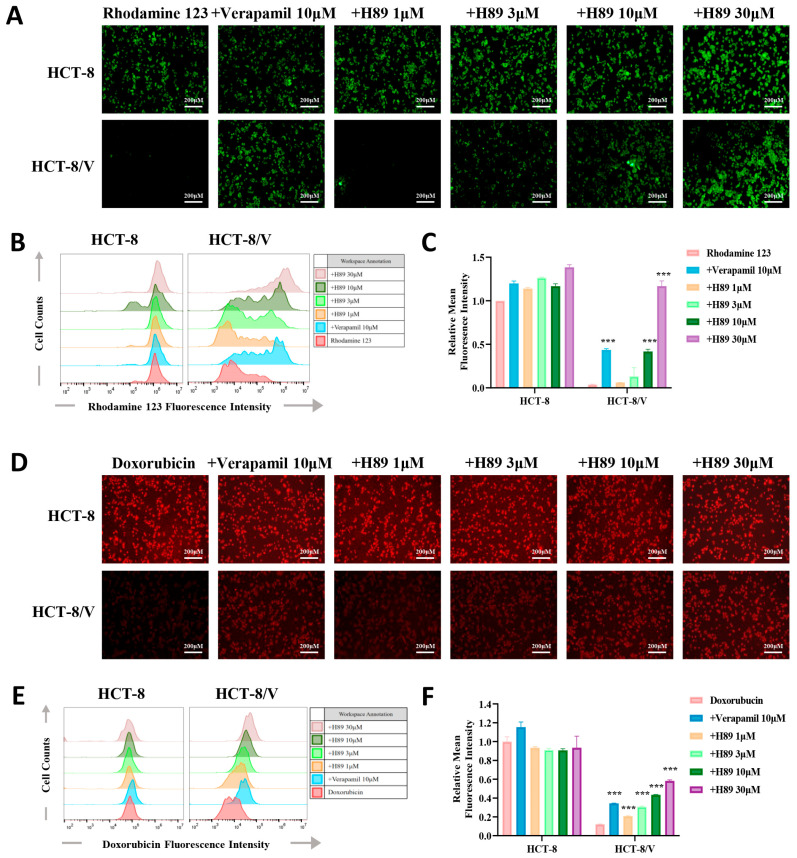
H89 enhances the intracellular accumulation of substrate drugs within HCT-8/V cells. After preincubatingHCT-8 and HCT-8/V cells with different concentrations of H89 or 10 μM verapamil for 1 h, 10 μM doxorubicin (red) or rhodamine 123 (green) was added and incubated together for 2 h. Subsequently, the cells were observed and imaged under a microscope, and quantitative analysis was performed using flow cytometry. Higher intracellular fluorescence intensity indicates a greater intracellular drug accumulation. Representative images (**A**,**D**), histograms (**B**,**E**), and quantitative data (**C**,**F**) are shown. The scale bar is 200 μm. “+” indicates the combination of two drugs. *** *p* < 0.001 compared to the corresponding control groups. Data are presented as the mean ± SD (*n* = 3).

**Figure 4 biomedicines-13-02869-f004:**
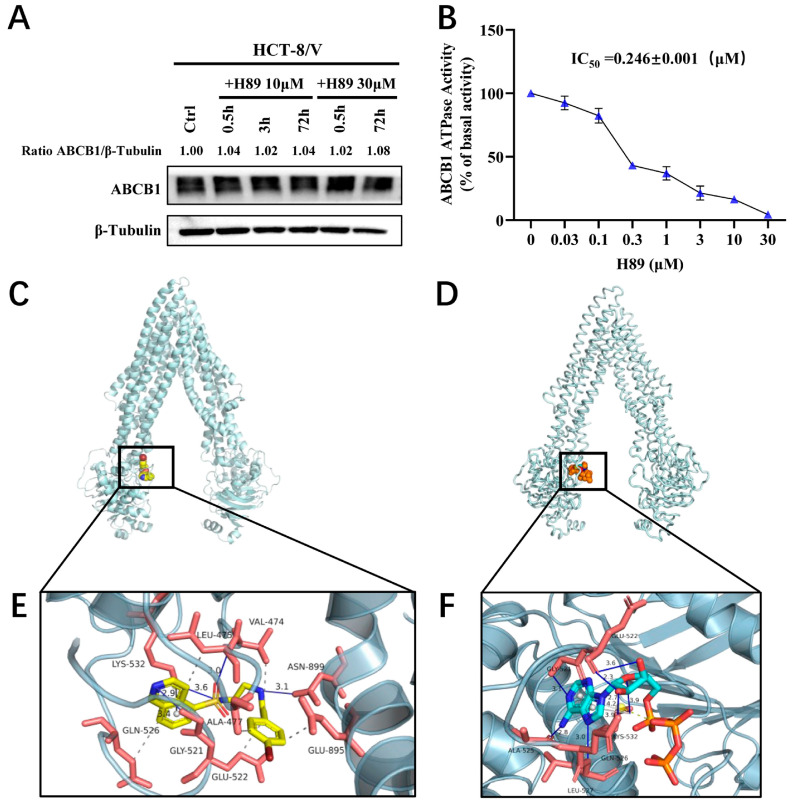
H89 inhibits the ATPase activity of ABCB1 and occupies its ATP-binding pocket. (**A**) ABCB1 expression levels in HCT-8/V cells treated with 10 μM and 30 μM H89 for the indicated time points were measured by Western blot. (**B**) The effect of H89 on the ATPase activity of ABCB1 was determined using the ABCB1 ATPase assay kit. Data are presented as the mean ± SD (*n* = 3). (**C**) The best-scoring pose of H89 in the ATP-binding domain of ABCB1. ABCB1 is displayed as a silver ribbon, and H89 is shown as a sphere model, predominantly in yellow, where yellow represents C (carbon), blue represents N (nitrogen), bright red represents O (oxygen), dark red represents Br (bromine), and tan represents S (sulfur). (**D**) The best-scoring binding conformation of ABCB1 with ATP. ABCB1 is displayed as a silver ribbon, and ATP is shown as a sphere model, predominantly in orange, where orange represents S (sulfur), blue represents N (nitrogen), cyan represents C (carbon), and bright red represents O (oxygen). (**E**) Detailed illustration of the interactions between H89 and the ATP-binding pocket of ABCB1. ABCB1 is displayed as a translucent silver ribbon, and the key amino acids are shown as orange-red sticks. H89 is presented as yellow sticks. Hydrophobic interactions are indicated by gray dashed lines, π-π stacking interactions by gray solid lineswith bond lengths of 2.9 Å, and hydrogen bonds by blue solid lines. The bond lengths between H89 and GLY-521, LYS-532, LEU-475, and ASN-899 are 3.4 Å, 3.6 Å, 3.0 Å, and 3.1 Å, respectively. (**F**) Detailed illustration of the interaction between ABCB1 and ATP. ABCB1 is displayed as a translucent silver ribbon, with key amino acids shown as orange-red sticks. ATP is presented as bright blue sticks. Gray dashed lines indicate hydrophobic interactions; yellow dashed lines represent salt bridges; gray solid lines denote π-π stacking interactions with bond lengths of 3.9 Å and 4.2 Å; blue solid lines indicate hydrogen bonds. The bond lengths between ATP and GLY-521, GLN-526, GLU-522, LYS-532, LEU-527, and ALA-525 are 3.1 Å, 2.7 Å, 2.3 Å, 3.9 Å, 3.0 Å, and 2.8 Å, respectively.

**Table 1 biomedicines-13-02869-t001:** Summary of the reversal effect of H89 on ABCB1-mediated MDR in CRC cells.

Compounds	IC_50_ (μM) ± SD (FoldResistance)	
	HCT-8	HCT-8/V
Doxorubicin	0.350 ± 0.081 (1.0)	6.978 ± 0.444 (19.94)
+Verapamil 10 μM	0.374 ± 0.002 (1.06)	0.934 ± 0.026 (2.67) **
+H89 3 μM	0.372 ± 0.160 (1.06)	1.586 ± 0.177 (4.53) **
+H89 10 μM	0.369 ± 0.183 (1.05)	0.648 ± 0.012 (1.85) **
Vincristine	0.025 ± 0.002 (1.0)	>30 (1200.0)
+Verapamil 10 μM	0.023 ± 0.001 (0.92)	0.238 ± 0.022 (9.52) **
+H89 3 μM	0.024 ± 0.001 (0.96)	0.140 ± 0.055 (5.60) **
+H89 10 μM	0.022 ± 0.001 (0.88)	0.047 ± 0.007 (1.88) **
Oxaliplatin	9.303 ± 0.549 (1.0)	10.101 ± 0.090 (1.08)
+Verapamil 10 μM	8.730 ± 0.441 (0.93)	7.892 ± 0.149 (0.84)
+H89 3 μM	8.924 ± 0.797 (0.96)	8.288 ± 0.092 (0.89)
+H89 10 μM	8.478 ± 0.475 (0.91)	7.351 ± 0.186 (0.79)

Foldresistance = IC_50_ value of HCT-8/V cells for each chemotherapeutic agent with and without inhibitors divided by IC_50_ value of HCT-8 cells against each chemotherapeutic drug without inhibitors. “+” indicates the combination of two drugs. ** *p* < 0.01 compared to the corresponding control groups. Data are presented as the mean ± SD (*n* = 3).

## Data Availability

The original contributions presented in this study are included in the article. Further inquiries can be directed to the corresponding authors.

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
