# Peer review of "H89 Reverses Multidrug Resistance in Colorectal Cancer by Inhibiting the ATPase Activity of ABCB1"

_biomedicines, 2025, doi:10.3390/biomedicines13122869_

Round 1
Reviewer 1 Report
Comments and Suggestions for Authors
The manuscript requires a comprehensive grammar and language check to enhance clarity and readability.
1. Figures 1–4 and their labels need to be clearer and must be prepared at a minimum resolution of 300 dpi as mandated by the journal.
2. The reference list should be meticulously reviewed to eliminate any duplicate or repeated citations.
3. The novelty of the study needs to be articulated more clearly in the introduction.
4. Details about the docking parameters and validation of the docking model are not sufficiently explained. The authors should specify the grid box size, software version, and any validation methods used, such as the re-docking of ATP and RMSD values, to ensure reliability.
5. Additionally, please discuss whether H89 may influence other kinases or pathways apart from ABCB1 within the manuscript.
6. Finally, the quality and labeling of the figures could be improved. For example, Figures 1-4 and 4C–F lack clear legends for color codes and atom labels. Moreover, the scale bars and indicators of statistical significance should be consistent across all figures.
7. Docking Methodology needs to be explained in detail to get more clearer view of the executed methodology.
Author Response
Comments 1: The manuscript requires a comprehensive grammar and language check to enhance clarity and readability.
Response 1: Thanks for the reviewer’s good suggestion. We have conducted a comprehensive grammar and language check to enhance clarity and readability.
Comments 2: Figures 1–4 and their labels need to be clearer and must be prepared at a minimum resolution of 300 dpi as mandated by the journal.
Response 2: Thanks for the reviewer’s nice suggestion. We have provided the clearer Figures 1-4 at a resolution of 300 dpi as mandated by the journal.
Comments 3: The reference list should be meticulously reviewed to eliminate any duplicate or repeated citations
Response 3: Thanks for the reviewer’s careful reading. We have checked and organized the list of references one by one.
Comments 4: The novelty of the study needs to be articulated more clearly in the introduction.
Response 4: Thanks for the reviewer’s nice suggestion. We have elaborated on the theoretical contributions of this study in the introduction, which enhances its innovativeness and academic value.
Comments 5: Details about the docking parameters and validation of the docking model are not sufficiently explained. The authors should specify the grid box size, software version, and any validation methods used, such as the re-docking of ATP and RMSD values, to ensure reliability
Response 5: We appreciate the reviewer for this kind suggestion. We have added information such as mesh box size and RMSD value in the molecular docking process to the methodology section.
Comments 6: Additionally, please discuss whether H89 may influence other kinases or pathways apart from ABCB1 within the manuscript
Response 6: Thanks for the reviewer’s nice suggestion. We have discussed whether H89 may influence other kinases or pathways apart from ABCB1.
Comments 7: Finally, the quality and labeling of the figures could be improved. For example, Figures 1-4 and 4C–F lack clear legends for color codes and atom labels. Moreover, the scale bars and indicators of statistical significance should be consistent across all figures.
Response 7: Thanks for the reviewer’s kind suggestion. We have improved the quality and labeling of the figures. Clear color-coded legends and atomic label explanations have been added to Figures 1-4 and 4C-F. The scale bars of all charts have standardized. *p<0.05, **p<0.01, and ***p<0.001 have been used consistently to ensure that all charts have the same format and complete information.
Comment 8. Docking Methodology needs to be explained in detail to get more clearer view of the executed methodology.
Response 8: Thanks for the reviewer’s kind suggestion. We have added the detailed docking methods to the methodology section.
Reviewer 2 Report
Comments and Suggestions for Authors
The authors investigate the potential of H89, a known protein kinase A (PKA) inhibitor, to reverse multidrug resistance (MDR) in colorectal cancer (CRC) by targeting the ABCB1 transporter. Using CRC cell models (HCT-8 and its resistant subline HCT-8/V), they demonstrate that H89 enhances drug sensitivity to ABCB1 substrates (doxorubicin, vincristine) without affecting non-substrates like oxaliplatin. Mechanistic assays reveal that H89 inhibits ABCB1 ATPase activity and increases intracellular accumulation of chemotherapeutic agents without changing ABCB1 protein expression. Docking analyses suggest H89 binds competitively to the ATP-binding pocket of ABCB1. The study’s novelty lies in identifying H89 as a dual-acting molecule capable of reversing ABCB1-mediated MDR through ATPase inhibition. The findings are relevant, as ABCB1-driven resistance remains a barrier in CRC therapy, and few agents have reached clinical translation. Some comments are listed below for authros attention
- The evidence that H89 inhibits ABCB1 ATPase activity is convincing but lacks confirmatory biochemical validation beyond the commercial assay kit. A kinetic or mutational study targeting the ATP-binding residues (e.g., Lys532 or Gly521) would strengthen the conclusion that inhibition is competitive and not allosteric.
- The molecular docking results are well-described, yet the absence of molecular dynamics (MD) simulations limits confidence in the stability of the proposed binding mode. A short MD simulation or binding energy decomposition would add robustness.
- As H89 inhibits multiple kinases, its selectivity toward ABCB1 should be clarified. The authors should discuss or experimentally test whether similar reversal effects occur in cells overexpressing other ABC transporters (ABCG2, ABCC1). This is essential to confirm target specificity.
- The study mentions minimal cytotoxicity at ≤30 μM, but additional controls (e.g., apoptosis or cell viability data beyond MTT) would provide a clearer toxicity profile.
- While Student’s t-test is used, some comparisons involve multiple groups (e.g., Figure 1C, Figure 3). An ANOVA with post hoc tests would be statistically more appropriate.
- The number of biological replicates (n=3) is adequate for exploratory work but marginal for publication-level mechanistic claims. Consider clarifying whether “n” refers to independent experiments or technical replicates.
- The authors claim that H89 is a “novel ABCB1-selective inhibitor.” Given H89’s known kinase promiscuity, this wording should be moderated. The data support an additional ABCB1-inhibitory property rather than selectivity.
- The discussion could better integrate previous studies where kinase inhibitors (e.g., staurosporine analogs) incidentally reversed MDR, situating H89 within this context.
- Table 1:
- Excellent quantitative summary. Include p-values for key comparisons (e.g., fold resistance changes).
- Consider merging redundant rows or adding a column showing “% reversal efficiency” to visualize H89’s potency relative to verapamil.
- Figure 1: Suggest adding representative microscopic images or viability staining (Trypan Blue or Calcein AM) to support MTT results.
- Figure 3: The fluorescence micrographs are central to the claim. Include a quantification panel (mean fluorescence intensity ± SD) directly on the figure for easier visual interpretation.
- A merged color image (overlaying doxorubicin fluorescence with DAPI) would further clarify subcellular localization.
- Figure 4: Currently text-heavy. Combine panels (C–F) into a single composite figure with improved annotation. Label interacting residues clearly (e.g., Gly521, Lys532) and include the docking score in the legend.
- Suggest adding an inset showing the ATP-binding site comparison between ATP and H89.
- Proposed Additions: Include a schematic figure summarizing the mechanism of H89 action (inhibition of ABCB1 ATPase leading to drug retention and MDR reversal). This would visually integrate the biochemical and cellular findings.
- Add a supplementary table listing key docking parameters (software version, grid size, number of runs, RMSD).
- Grammar and syntax are generally strong but some sentences in the Discussion are overly long. Simplify to improve readability.
- Standardize figure legends for consistency (e.g., “Data are mean ± SD, n=3, *p<0.05”).
- Verify the uniformity of chemical names and concentrations.
- Include catalog numbers and suppliers for all antibodies in the Methods for full reproducibility.
- The phrase “did not alter the expression of ABCB1” should cite the specific Western blot quantification (relative to β-tubulin) to confirm no subtle downregulation.
- Minor typographical corrections: “inhibit its ATPaseactivity reaction” should read “inhibit its ATPase activity reaction.”
Author Response
Comments 1: The evidence that H89 inhibits ABCB1 ATPase activity is convincing but lacks confirmatory biochemical validation beyond the commercial assay kit. A kinetic or mutational study targeting the ATP-binding residues (e.g., Lys532 or Gly521) would strengthen the conclusion that inhibition is competitive and not allosteric.
Response 1: We agree with the reviewer. A kinetic or mutational study targeting the ATP-binding residues (e.g., Lys532 or Gly521) will indeed strengthen the conclusion that inhibition is competitive and not allosteric.
Comments 2: The molecular docking results are well-described, yet the absence of molecular dynamics (MD) simulations limits confidence in the stability of the proposed binding mode. A short MD simulation or binding energy decomposition would add robustness.
Response 2: We agree with the reviewer. A short MD simulation or binding energy decomposition will indeed add robustness.
Comments 3: As H89 inhibits multiple kinases, its selectivity toward ABCB1 should be clarified. The authors should discuss or experimentally test whether similar reversal effects occur in cells overexpressing other ABC transporters (ABCG2, ABCC1). This is essential to confirm target specificity.
Response 3: Thanks for the reviewer’s good suggestion. We have discussed the selectivity of H89 to clarify its targeting specificity towards ABCB1.
Comments 4: The study mentions minimal cytotoxicity at ≤30 μM, but additional controls (e.g., apoptosis or cell viability data beyond MTT) would provide a clearer toxicity profile.
Response 4: Thanks for the reviewer’s nice suggestion. Our data of figure 2 showed that H89 at 10 μM did not alert the cell cycle distribution, especially sub G1 which indicates cells apoptosis.
Comments 5: While Student’s t-test is used, some comparisons involve multiple groups (e.g., Figure 1C, Figure 3). An ANOVA with post hoc tests would be statistically more appropriate.
Response 5: We appreciate the reviewer for this great suggestion. We have added One-Way ANOVA statistical method to the methodology section.
Comments 6: The number of biological replicates (n=3) is adequate for exploratory work but marginal for publication-level mechanistic claims. Consider clarifying whether “n” refers to independent experiments or technical replicates.
Response 6: Thanks for the reviewer’s good suggestion. “n” refers to independent experiments in our data.
Comments 7: The authors claim that H89 is a “novel ABCB1-selective inhibitor.” Given H89’s known kinase promiscuity, this wording should be moderated. The data support an additional ABCB1-inhibitory property rather than selectivity.
Response 7: We agree with the reviewer. We have corrected this in the revised manuscript.
Comments 8: The discussion could better integrate previous studies where kinase inhibitors (e.g., staurosporine analogs) incidentally reversed MDR, situating H89 within this context.
Response 8: We appreciate the reviewer for this good suggestion. We have integrated previous studies where kinase inhibitors (e.g., staurosporine analogs) incidentally reversed MDR, situating H89 within this context in the disccusion.
Comments 9: Table 1:Excellent quantitative summary. Include p-values for key comparisons (e.g., fold resistance changes). Consider merging redundant rows or adding a column showing “% reversal efficiency” to visualize H89’s potency relative to verapamil.
Response 9: We appreciate the reviewer for this great suggestion. We have supplemented the p-values in Table 1 with **p< 0.01. For reversal efficiency, we have already indicated the fold-resistance, which is a straightforward indicator.
Comments 10: Figure 1: Suggest adding representative microscopic images or viability staining (Trypan Blue or Calcein AM) to support MTT results.
Response 10: We agree with the reviewer. The representative microscopic images or viability staining (Trypan Blue or Calcein AM) will indeed support MTT results.
Comment 11. Figure 3: The fluorescence micrographs are central to the claim. Include a quantification panel (mean fluorescence intensity ± SD) directly on the figure for easier visual interpretation. A merged color image (overlaying doxorubicin fluorescence with DAPI) would further clarify subcellular localization.
Response 11: We agree with the reviewer. Because our microscope was unable to obtain quantitative fluorescence intensity, we performed the quantitative analysis of intracellular fluorescence using flow cytometry. DAPI staining can indeed better reveal subcellular localization.
Comments 12: Figure 4: Currently text-heavy. Combine panels (C–F) into a single composite figure with improved annotation. Label interacting residues clearly (e.g., Gly521, Lys532) and include the docking score in the legend. Suggest adding an inset showing the ATP-binding site comparison between ATP and H89.
Response 12: Thanks for the reviewer’s kind suggestion. We retained the CE diagram and replaced the DF diagram with a docking illustration of ATP and ABCB1 to clearly highlight the contrasting details between H89 and ATP. We also revised the figure captions for clearer descriptions.
Comments 13: Include a schematic figure summarizing the mechanism of H89 action (inhibition of ABCB1 ATPase leading to drug retention and MDR reversal). This would visually integrate the biochemical and cellular findings.
Response 13: Thanks for the reviewer’s good suggestion. We have provided a detailed mechanism diagram in the Abstract Figure file.
Comments 14: Add a supplementary table listing key docking parameters (software version, grid size, number of runs, RMSD).
Response 14: Thanks for the reviewer’s kind suggestion. We have added the key docking parameters (software version, grid size, number of runs, RMSD) to the methodology section.
Comments 15: Grammar and syntax are generally strong but some sentences in the Discussion are overly long. Simplify to improve readability.
Response 15: Thanks for the reviewer’s careful reading. We have simplified overly long sentences in the discussion section to improve readability.
Comments 16: Standardize figure legends for consistency (e.g., “Data are mean ± SD, n=3, *p<0.05”).
Response 16: Thanks for the reviewer’s nice suggestion. We have standardized figure legends for consistency (e.g., “Data are mean ± SD, n=3, *p<0.05”).
Comments 17: Verify the uniformity of chemical names and concentrations.
Response 17: We thank the reviewer for careful reading. We have verified the the uniformity of chemical names and concentrations.
Comments 18: Include catalog numbers and suppliers for all antibodies in the Methods for full reproducibility.
Response 18: We thank the reviewer for careful reading. We have included catalog numbers and suppliers for all antibodies in the Methods
Comments 19: The phrase “did not alter the expression of ABCB1” should cite the specific Western blot quantification (relative to β-tubulin) to confirm no subtle downregulation.
Response 19: Thanks for the reviewer’s kind suggestion. We have cited the specific Western blot quantification (relative to β-tubulin).
Comments 20: Minor typographical corrections: “inhibit its ATPaseactivity reaction” should read “inhibit its ATPase activity reaction.”
Responses 20: We thank the reviewer for careful reading. We have corrected the typographical errors.
Reviewer 3 Report
Comments and Suggestions for Authors
The article refers to the effect of H89, a PKA inhibitor, on treating the drug-resistant colon cancer cell line HCT-8/V and comparing it to its counterpart, HCT-8. The general rationale is adequate, although not well expressed in the introduction. The abstract is partially ambiguous since it does not refer to the results obtained, but rather concludes and proposes the mechanism. The methodology needs improvement since it lacks adequate controls in cell cycle analysis, and the ABCB1-mediated MDR in CRC cells (Verapamil should have also been used in this study). Please consider ordering the methodology so the reader can understand the rationale of the different experiments. Additionally, have the authors considered performing a time kinetics study with the PKA inhibitor to determine the optimal effect? H89 can also affect glucose metabolism, which plays a crucial role in drug resistance; however, there are no studies on that topic.
Please use a better format for the gels, TIFF, or PEG. In Table 1, how did the authors calculate the fold resistance of vincristine with an IC50>30? Figure 2A: Please determine the amount of apoptotic DNA in the figure. The program used can give the values. Part B of the figure must be enhanced. The legend of Figure 3 is very poor. I do not understand what do you mean by doxorubicin or rhodamine 123. Since cells are pretreated, order the Figure accordingly. Figure 4, part B, should be better represented, and the calculation of IC50 must be stated. Graph Pad, I assume, in which case a proper curve can be drawn. I am however, not convinced that an n=3 is enough. Do you think your result are similar to those of cAMPS-Rp, triethylammonium salt?
The discussion may be enhanced. Based on the complexity of the platinum-based drugs in cell culture, have the authors analyzed the possibility of using a delivery system that includes H89 along with the platinum drugs? Drug efficiency may be highly enhanced.
Author Response
Comments 1: The general rationale is adequate, although not well expressed in the introduction.
Response 1: We thank the reviewer for careful reading. We have revised the introduction to improve the general rationale.
Comments 2: The abstract is partially ambiguous since it does not refer to the results obtained, but rather concludes and proposes the mechanism.
Response 2: We thank the reviewer for careful reading. We have added the results in the revised abstract.
Comments 3: The methodology needs improvement since it lacks adequate controls in cell cycle analysis, and the ABCB1-mediated MDR in CRC cells (Verapamil should have also been used in this study).
Response 3: We agree with the reviewer. Verapamil as a control in cell cycle analysis will indeed improve the results.
Comments 4: Please consider ordering the methodology so the reader can understand the rationale of the different experiments.
Response 4: We thank the reviewer for careful reading. We have reorganized the methodology so the reader can understand the rationale of the different experiments.
Comments 5: have the authors considered performing a time kinetics study with the PKA inhibitor to determine the optimal effect? H89 can also affect glucose metabolism, which plays a crucial role in drug resistance; however, there are no studies on that topic.
Response 5: Thanks for the reviewer’s kind suggestion. A time kinetics study with the PKA inhibitor will indeed determine the optimal effect. We did not study glucose metabolism, but glucose metabolism affected by H89 plays a crucial role in drug resistance.
Comments 6: Please use a better format for the gels, TIFF, or PEG.
Response 6: We thank the reviewer for careful reading. We have used a better TIFF format for the gels.
Comments 7: In Table 1, how did the authors calculate the fold resistance of vincristine with an IC50>30?
Response 7: We used the IC50 value, which is "30/0.025", for calculation, so the result is displayed as ">1200".
Comments 8: Figure 2A: Please determine the amount of apoptotic DNA in the figure. The program used can give the values. Part B of the figure must be enhanced.
Response 8: We appreciate the reviewer for this nice suggestion. We have determined the amount of apoptotic DNA as subG1 phase in the Figure 2A and enhanced the Figure 2B.
Comments 9: The legend of Figure 3 is very poor. I do not understand what do you mean by doxorubicin or rhodamine 123. Since cells are pretreated, order the Figure accordingly.
Response 9: We thank the reviewer for careful reading. We have revised the legend of Figure 3. The specific experimental steps were as follows: First, parental cells and drug-resistant cells were set up as control, different concentrations of H89, and 10 μM verapamil pretreatment groups. After pretreatment for 1 hour with H89 or verapamil, 10 μM doxorubicin (red) or 10 μM rhodamine 123 (green) was added to all pretreatment groups, and they were incubated together for 2 hours.
Comments 10: Figure 4, part B, should be better represented, and the calculation of IC50 must be stated. Graph Pad, I assume, in which case a proper curve can be drawn. I am however, not convinced that an n=3 is enough.
Response 10: We thank the reviewer for careful reading. A proper curve can be drawn with Graph Pad. Our IC50 value was calculated through dose-response curve simulation.
Comments 11: Do you think your result are similar to those of cAMPS-Rp, triethylammonium salt?
Response 11: Both cAMPS-Rp and H89 are PKA inhibitors, but their chemical structures are different. Our results show that H89 inhibits the ATPase activity of ABCB1. Whether cAMPS-Rp also inhibits the ATPase activity of ABCB1 needs to be investigated.
Comments 12: The discussion may be enhanced. Based on the complexity of the platinum-based drugs in cell culture, have the authors analyzed the possibility of using a delivery system that includes H89 along with the platinum drugs? Drug efficiency may be highly enhanced.
Response 12: Thanks for the reviewer’s nice suggestion. We have enhanced the discussion to analyze the possibility of using a delivery system that includes H89 along with the platinum drugs.
Round 2
Reviewer 2 Report
Comments and Suggestions for Authors
Thank you, No further comments
Reviewer 3 Report
Comments and Suggestions for Authors
The manuscript was improved. Although the gels in the supplementary file were not modified as requested, I suppose the authors will do it. I do not have any other comment